# Iron Metabolism and Inflammatory Mediators in Patients with Renal Dysfunction

**DOI:** 10.3390/ijms25073745

**Published:** 2024-03-27

**Authors:** Tomomi Matsuoka, Masanori Abe, Hiroki Kobayashi

**Affiliations:** Division of Nephrology, Hypertension and Endocrinology, Department of Medicine, Nihon University School of Medicine, Tokyo 173-8610, Japan

**Keywords:** chronic kidney disease, iron metabolism, uremic toxins, inflammatory mediators, hepcidin

## Abstract

Chronic kidney disease (CKD) affects around 850 million people worldwide, posing significant challenges in healthcare due to complications like renal anemia, end-stage kidney disease, and cardiovascular diseases. This review focuses on the intricate interplay between iron metabolism, inflammation, and renal dysfunction in CKD. Renal anemia, prevalent in CKD, arises primarily from diminished erythropoietin (EPO) production and iron dysregulation, which worsens with disease progression. Functional and absolute iron deficiencies due to impaired absorption and chronic inflammation are key factors exacerbating erythropoiesis. A notable aspect of CKD is the accumulation of uremic toxins, such as indoxyl sulfate (IS), which hinder iron metabolism and worsen anemia. These toxins directly affect renal EPO synthesis and contribute to renal hypoxia, thus playing a critical role in the pathophysiology of renal anemia. Inflammatory cytokines, especially TNF-α and IL-6, further exacerbate CKD progression and disrupt iron homeostasis, thereby influencing anemia severity. Treatment approaches have evolved to address both iron and EPO deficiencies, with emerging therapies targeting hepcidin and employing hypoxia-inducible factor (HIF) stabilizers showing potential. This review underscores the importance of integrated treatment strategies in CKD, focusing on the complex relationship between iron metabolism, inflammation, and renal dysfunction to improve patient outcomes.

## 1. Introduction

Chronic kidney disease (CKD) represents a significant global health concern, affecting an estimated 850 million individuals worldwide [1]. Its prevalence continues to rise, positioning it among the most prevalent diseases internationally. CKD, characterized by persistent renal damage over a period exceeding three months or a glomerular filtration rate (GFR) under 60 mL/min/1.73 m^2^ for a similar duration, was initially defined by the Kidney Disease Outcomes Quality Initiative (KDOQI) in 2002 and later revised by the Kidney Disease Improving Global Outcomes (KDIGO) [2,3]. The etiology of CKD is diverse, encompassing diabetes, hypertension, chronic glomerulonephritis, genetic disorders, chronic pyelonephritis, autoimmune diseases, polycystic kidney disease, and medication-induced damage. As CKD progresses, it leads to the accumulation of metabolic byproducts, typically excreted by the kidneys, resulting in further damage to various organs and cells. The progression of CKD ultimately leads to end-stage kidney disease (ESKD), necessitating interventions like hemodialysis or kidney transplantation. CKD also elevates the risk of cardiovascular events (CVE), often resulting in heart failure. Notably, cardiovascular complications are the leading cause of death in CKD patients, often preceding the onset of ESKD [4]. Additionally, anemia associated with CKD is linked to diminished quality of life (QOL), increased hospitalizations, cognitive impairments, and heightened mortality risk [5].

## 2. Renal Anemia

### 2.1. Concept of Renal Anemia

Renal anemia, a significant complication of CKD, results from a decline in renal function, which leads to diminished erythropoietin (EPO) synthesis and, consequently, inadequate maintenance of hemoglobin (Hb) levels. Clinically, it is defined by Hb concentrations below 13 g/dL in men and below 12 g/dL in women [6]. The prevalence of renal anemia proportionately increases with the progression of CKD, evident from prevalence rates of 17.4% in stage 3 (eGFR 30–59 mL/min/1.73 m^2^), escalating to 50.3% in stage 4 (eGFR 15–29 mL/min/1.73 m^2^), and peaking at 53.4% in stage 5 (eGFR < 15 mL/min/1.73 m^2^) [7].

### 2.2. Erythropoietin as a Central Factor in Erythropoiesis

EPO is a critical glycoprotein hormone for erythropoiesis, and its expression increases in hypoxic and anemic conditions. It circulates in the plasma and stimulates marrow progenitors, thereby increasing red cell production [8]. It achieves this by binding to EPO receptors (EPORs) on erythroid progenitor cells in the bone marrow, thereby inhibiting apoptosis and stimulating both proliferation and differentiation [9]. During late fetal development, erythropoiesis is supported by EPO production from hepatocytes, but in adults, EPO production primarily occurs in the kidneys. EPO is produced by renal erythropoietin-producing (REP) cells in the proximal peritubular interstitium of the kidney [10]. Furthermore, it has been shown that REP cells originate from myelin protein zero cells in the neural crest [11]. As CKD progresses, REP cells undergo a phenotypic transition to myofibroblasts, during which they lose their ability to synthesize EPO [12].

The main determinant of EPO production is the transcriptional activity of its gene in the kidneys, which is related to local oxygen tensions. Thus, EPO production is regulated by hypoxia-inducible factors (HIFs). HIF is a transcription factor consisting of a dimer of HIF-α and HIF-β. HIF-α has three isoforms: HIF-1α, HIF-2α, and HIF-3α. In the kidney, HIF-1α is mainly located in tubular epithelial cells, while HIF-2α is found in endothelial cells and fibroblasts [13]. Under normal oxygen conditions, HIF-1α and HIF-2α undergo hydroxylation of proline residues by prolyl hydroxylases (PHD), leading to ubiquitination by the von Hippel-Lindau (VHL) protein and subsequent degradation by the proteasome. On the other hand, under hypoxic conditions, reduced activity of PHD stabilizes HIF-α proteins, forming a dimer with HIF-β and binding to hypoxia response elements (HREs), inducing the expression of target genes such as EPO (Figure 1).

Although the kidney is the primary site of EPO production associated with HIF activation, EPO production also originates from the liver, as low levels of EPO production were observed with PHD inhibitors in CKD patients after nephrectomy [14]. PHD also has three isoforms, and while in the liver, all PHD isoforms must be inhibited [15,16], in the kidney, PHD2 is said to be most involved [17]. HIF activation influences not only EPO production but also genes that play important roles in iron metabolism, including transferrin (Tf) [18], ferroportin (FPN) [19], and possibly hepcidin [19].

### 2.3. Mechanism of Renal Anemia

In CKD, a critical balance between oxygen supply and demand around EPO-producing cells profoundly influences anemia development. Oxygen supply, determined by factors like renal blood flow and Hb concentration, is juxtaposed against oxygen consumption, predominantly driven by sodium reabsorption in the renal tubules [20]. In CKD, diminished renal blood flow may lead to inadequate oxygen delivery to tissues, while concomitant tubular damage can lower local oxygen consumption. This complex interplay can result in insufficient stimulation of EPO production, even in the presence of hypoxia or anemia. Consequently, this dysregulation can lead to persistent anemia, a hallmark of CKD, stemming from a relative deficiency in EPO production, a key aspect of the pathophysiology of renal anemia.

CKD patients often exhibit elevated levels of uremic toxins in circulation. These toxins, including compounds like blood urea nitrogen (BUN), and guanidine compounds, are thought to potentially suppress red blood cell production [21]. Some researchers have pointed to increased levels of inflammatory cytokines, such as interferon and tumor necrosis factor-alpha (TNF-α), commonly observed in CKD, which may lead to a decreased sensitivity of erythroid progenitors to EPO [22,23]. This observation suggests a potential connection between cytokine dysregulation and anemia in CKD.

Iron deficiency, both absolute and functional, plays a pivotal role in CKD-related anemia. Absolute iron deficiency refers to a depletion of iron stores, whereas functional iron deficiency denotes impaired utilization of available iron. Factors contributing to these deficiencies in CKD include chronic blood loss, reduced iron absorption, and ongoing inflammation. A comprehensive understanding of these factors is crucial for effective anemia management in CKD, emphasizing the need for a dual approach that addresses both EPO and iron deficiency.

## 3. Iron Metabolism in Chronic Kidney Disease

### 3.1. Physiological Role and Homeostasis of Iron

Iron, a vital trace element, participates in oxygen reactions, producing reactive oxygen species (ROS) like hydroxyl radicals and hydrogen peroxide. It is involved in Hb synthesis in red blood cells, oxygen transport, various redox reactions, cell proliferation, and anti-inflammatory actions. However, excess iron can cause tissue damage due to oxidative stress [24]. To maintain a critical balance in iron homeostasis, iron metabolism is regulated by various iron metabolic proteins [25,26].

The human body contains approximately 3–5 g of iron, with 60% in Hb in red blood cells, 30% stored in hepatocytes and reticuloendothelial macrophages, and 10% in muscle myoglobin. The body loses about 1–2 mg of iron daily through sweat, bleeding, and the desquamation of intestinal epithelial cells, compensated by iron absorption from the duodenum [27,28]. Daily iron losses are replenished by the absorption of approximately 1–2 mg of dietary iron from the duodenum, but Hb synthesis alone requires 20–25 mg of iron per day. To support Hb synthesis and other metabolic processes, iron recycling and strict regulation within the body are necessary. Iron supply in the blood is largely met through recycling from reticuloendothelial macrophages and replenishment from hepatocytes.

### 3.2. Absorption, Utilization, and Recycling of Iron

Iron absorption occurs in the duodenum. Dietary iron exists as heme and non-heme iron. Non-heme iron, typically in the form of ferric iron (Fe^3+^), is reduced to ferrous iron (Fe^2+^) by duodenal cytochrome B [29] and transported into enterocytes by divalent metal transporter 1 (DMT1) [30,31,32]. Heme iron is broken down into Fe^2+^ and biliverdin by heme oxygenases (HO)-1 and HO-2 [33,34,35]. Inside enterocytes, Fe^2+^ is stored in ferritin, an iron storage protein.

Absorbed Fe^2+^ is exported into the plasma by the iron exporter FPN [27,36,37], where it binds to Tf for circulation [38,39,40]. Circulating Tf-bound iron is taken up by cells and tissues via Tf receptor 1 (TFR1) [41,42], transported to the liver and spleen for storage in ferritin, or to the bone marrow for erythropoiesis [43]. The main consumer of iron is bone marrow, which requires significant iron for Hb production in red blood cells. Aged red blood cells are phagocytized mainly in the spleen, with storage iron replenished by macrophages recycling iron from erythrocyte destruction. Additionally, EPO induces red blood cell production, replenishing iron through bone marrow recycling [44] [Figure 2].

### 3.3. Factors Involved in Iron Regulation and Control

Aged red blood cells are recycled by macrophages through phagocytosis, and then HO-1 breaks down heme iron to Fe^2+^, which, like in enterocytes, is released into the blood via FPN [27,36,37]. FPN1 expression is regulated by hepcidin, a peptide hormone synthesized mainly in the liver. The liver, besides being a major storage site for iron, emphasizes its role in iron homeostasis [27,39]. Hepcidin regulates the uptake of iron from the intestine and the release of stored iron [45] (Figure 2).

### 3.4. Iron Metabolism in the Kidneys

The kidneys, which are abundant in mitochondria containing heme iron, play a vital role in iron metabolism. Iron filtered through the glomeruli is reabsorbed in the renal tubules [46,47]. The kidneys use TFR1, expressed in cortical and medullary cells, to absorb Tf-bound iron (TBI) [48]. In proximal tubular cells, megalin synthesis impairment in a mouse model, leading to excessive Tf excretion and the inability to import Tf, highlights the importance of megalin-mediated TBI uptake in the kidneys [49]. Iron bound to L-ferritin is absorbed through scavenger receptor class A member 5 in the renal interstitium [50,51].

Along the nephron, iron is reabsorbed using other transporters like DMT1, ZIP8, and ZIP14, competing with other divalent metals like copper and manganese [52]. ZIP8 and ZIP14, initially identified as zinc transporters and expressed in the proximal tubules, can transport non-TBI (NTBI), cadmium, and manganese [53]. In addition to these iron transporters, other receptors/transporters assist in renal iron reabsorption [54]. Hb released from erythrocytes during hemolysis is filtered by the kidneys and internalized in the proximal tubules via megalin/cubilin receptors [55,56]. Heme is then catabolized by heme oxygenase and stored in conjunction with ferritin. The renal parenchyma absorbs both TBI and NTBI, with the ferritin H subunit showing ferroxidase activity to safely store iron by converting Fe^2+^ to Fe^3+^ in the ferritin mineral core [57]. Mouse proximal tubules express high levels of iron regulatory protein 1, active under physiological oxygen concentrations [48].

### 3.5. Renal Iron Regulatory Factors: Ferroportin and Hepcidin

In the kidneys, iron is actively exported through FPN1, the only known iron export protein. FPN1 is primarily located in the brush border membrane of the apical side, with weaker expression in the basolateral membrane of proximal tubular cells and cytoplasmic compartments [58,59]. In conditions of iron overload, FPN1 is mainly highly expressed in the apical side of the proximal tubules, suggesting an attempt to excrete excess iron into the tubular lumen. FPN1 is involved in the movement of iron within and outside the renal parenchyma, but hepcidin regulates the systemic export of iron through FPN1 [60]. Hepcidin binds to FPN, inducing its internalization, ubiquitination, and degradation, thereby limiting the release of iron into circulation and its utilization in target tissues [61,62]. Excess hepcidin is excreted by the kidneys [28,61].

Hepcidin mRNA detected not only in the cortex but also in the renal medulla suggests local synthesis in the kidney [63]. Kulaksiz et al. found high expression of hepcidin in the thick ascending limb and connecting tubules of the renal cortex and moderate expression in the medullary thick ascending limb and collecting ducts in rats. Hepcidin was localized to the apical cell pole of renal epithelial cells [64].

A previous study has shown that hepcidin knockout mice exhibit increased expression of renal FPN and ferritin, and the administration of exogenous hepcidin to cultured kidney cells reduced DMT1 protein expression, indicating that hepcidin decreases iron transport [65]. Post-unilateral ureteral obstruction, high hepcidin expression was observed in the kidney, with decreased FPN1 expression, indicating an inverse correlation between hepcidin and FPN levels [66]. Mohammad et al., using mouse model with an inducible kidney-tubule specific knock-in of fpnC326Y, which encodes a hepcidin-resistant FPN termed FPNC326Y, showed that under physiological conditions, endogenous hepcidin controls tubular iron reabsorption via FPN [67]. These mice consistently showed reduced renal iron under normal iron availability conditions but transiently increased systemic iron indices. Under iron overload conditions, these mice had mild renal iron overload but significant systemic iron overload compared to control mice. Furthermore, wild-type mice (with an intact renal HAMP/FPN axis) developed renal iron overload under the same systemic iron overload conditions, whereas hemochromatosis mice (with a disrupted renal HAMP/FPN axis) did not. This finding underscores the importance of the renal HAMP/FPN axis in managing iron homeostasis in both the kidney and the body during iron overload.

Hepcidin production is stimulated by increased iron intake, inflammatory states, and infection, while it is inhibited by iron deficiency and hypoxia. It is also regulated by erythropoiesis and endocrine signals [28,44,68,69,70,71]. Hematopoietic stimulation by EPO increases erythroferon secreted by erythroblasts and decreases hepcidin secretion [72]. Furthermore, HIFs, through the stimulation of EPO-induced erythropoiesis, indirectly reduce the serum levels of hepcidin [73]. Hepcidin, induced by inflammation, particularly interleukin-6 (IL-6), has been identified as an antimicrobial peptide [74]. In conditions of iron overload, the expression of hepcidin is induced by the activation of the bone morphogenetic protein (BMP)/SMAD pathway (the human homolog of Drosophila mothers against decapentaplegic). Hemojuvelin (HJV), a membrane protein, is implicated in the iron overload condition known as juvenile hemochromatosis. Serving as a BMP co-receptor, HJV signals through the SMAD pathway to regulate hepcidin expression [75]. Although the study by Styczynski et al. did not focus on patients with CKD, it revealed that soluble HJV (sHJV) plays a role in the regulation of iron metabolism. Furthermore, sHJV could be a valuable biomarker for assessing short-term outcomes in pediatric patients after hematopoietic cell transplantation or chemotherapy, both of which are contexts where iron overload is a concern [76].

In this way, iron metabolism proteins regulate iron homeostasis, maintaining the body’s iron balance. Especially, hepcidin plays a crucial role in iron control. CKD, characterized by an accumulation of uremic toxins and inflammatory cytokines, leads to a chronic inflammatory state, affecting iron metabolism. The expression of hepcidin, an iron regulatory factor, is also influenced by uremic toxins and inflammatory cytokines.

## 4. Uremic Toxins and Their Impact on Renal Anemia and Iron Metabolism

### 4.1. Uremic Toxins

As CKD progresses, the accumulation of uremic toxins occurs within the body. These toxins affect various cell types, including vascular endothelial cells, mesangial cells, and tubular epithelial cells, leading to oxidative stress, chronic inflammation, and a reduction in cell proliferation. The European Uremic Toxin (EUTox) Working Group has extensively listed 146 known uremic toxins, categorizing them based on their molecular weight, protein-binding capacity, and their removal patterns during dialysis into three distinct groups [77,78,79,80,81]:

Water-soluble low-molecular-weight compounds: These are molecules with a molecular weight of less than 500 Daltons. Common examples include urea, uric acid, guanidine, phenylacetic acid, and neopterin [80,82].

Middle molecules: These are larger, with a molecular weight exceeding 500 Daltons, and include β2-microglobulin, parathyroid hormone, fibroblast growth factor 23 (FGF23), cystatin C, leptin, and adiponectin [80,82].

Protein-bound low-molecular-weight compounds: They are generally small in size (less than 500 Daltons), but are tightly bound to proteins. This category includes compounds like indoxyl sulfate (IS), p-cresyl sulfate (PCS), furancarboxylic acid, hippuric acid, and kynurenic acid [80,82,83].

Unlike the water-soluble low-molecular-weight compounds, which are easily removed through standard dialysis, middle molecules require a dialyzer membrane with larger pores for their removal [84,85]. In contrast, protein-bound low-molecular-weight compounds, due to their strong binding with abundant serum proteins like albumin, are not readily removed by current dialysis techniques, regardless of membrane pore size [86,87].

### 4.2. Indoxyl Sulfate: A Representative Uremic Toxin

Among protein-bound low-molecular-weight compounds, IS stands out as one of the most studied toxins, primarily because it is not effectively removed by conventional dialysis methods. IS is generated when tryptophan, an amino acid found in the diet, is converted into indole by intestinal bacteria. This indole is then transported to the liver, where it undergoes sulfation, mediated by cytochrome P450 enzymes (specifically CYP2A6/2E1) and the sulfotransferase enzyme (SULT1A1), transforming into IS [88].

Once formed, IS is transported into renal tubular epithelial cells through organic anion transporters (OATs), located on the basolateral membrane of the proximal tubules in the kidneys. It is then excreted into the urine [89,90]. However, in patients with impaired kidney function, the excretion of IS decreases, leading to its accumulation in the circulation [91]. In CKD mice deficient in intestinal microbiota under germ-free conditions, IS was not detected in plasma, indicating that IS is a metabolite dependent on intestinal bacterial metabolism [92]. A clinical study has shown that high levels of IS (≥6.124 mg/L) in patients with CKD are significantly associated with the progression of kidney disease towards dialysis dependency [93]. Due to its high protein-binding rate, especially with serum albumin (approximately 95%), IS is difficult to remove through dialysis [94].

### 4.3. Indoxyl Sulfate and Its Impact on Kidney Damage

IS is recognized as a nephrotoxic substance that exacerbates renal failure. It has been demonstrated that administration of indole or IS in uremic rats accelerated the progression of glomerulosclerosis, indicating its role in advancing renal failure [95]. IS is primarily responsible for tubular damage via oxidative and endoplasmic reticulum (ER) stress. It enhances the expression of transforming growth factor (TGF)-β1, tissue inhibitor of metalloproteinases 1, and proα1(I) collagen, contributing to interstitial fibrosis and glomerulosclerosis [96]. In human proximal tubule epithelial cells (HK-2), IS uptake through OAT leads to increased ROS production, activation of nuclear factor-kappa B (NF-κB), and elevation of PAI-1 expression [97]. IS, alongside PCS, activates the intrarenal Renin-Angiotensin-Aldosterone System/TGF-β pathway and induces epithelial-to-mesenchymal transition in tubular cells, further promoting renal fibrosis [98] (Figure 3). In patients with CKD, IS enhances ROS production in endothelial cells, activates NADPH oxidase, decreases levels of the active antioxidant glutathione (GSH), and induces oxidative stress [99]. It also reduces the activity of the renal antioxidant enzyme superoxide dismutase [100]. IS-related ER stress in tubular cells leads to an increase in activating transcription factor 4, inducing the expression of IL-6 and p21, thus inhibiting cell proliferation and hindering tubular cell regeneration after injury [101]. The accumulation of IS in CKD is a predictor of kidney function decline [102] and, like inflammation, induces ROS, adversely affecting various organs [103] (Figure 3).

### 4.4. The Impact of Indoxyl Sulfate on Iron Metabolism and Renal Anemia

IS not only accelerates the progression of CKD but also plays a significant role in iron metabolism and the development and regulation of renal anemia. Uremic toxins like IS, which accumulate in the blood instead of being filtered and excreted by the kidneys, lead to a deficiency in EPO production by renal interstitial fibroblasts (peritubular cells of the kidney), resulting in a decrease in red blood cell production and anemia [104,105].

In CKD, both HIF-1 and HIF-2 are activated, with HIF-2α being a crucial regulator of hypoxic EPO induction [106]. The absence of HIF-2α in renal tissues can lead to severe anemia [107]. The impact of IS is associated with impaired renal EPO synthesis due to the HIF-dependent suppression of EPO gene transcription and worsening renal hypoxia [108]. Independent studies have confirmed that IS inhibits hypoxic induction of HIF-1 target genes and disrupts the function of the C-terminal transactivation domain of HIF-1α in HK-2 [109] (Figure 4). IS acts as a ligand for the aryl hydrocarbon receptor (AhR), and its effects through AhR have been demonstrated [110]. Asai et al. showed that AhR activation plays a crucial role in IS-mediated suppression of HIF activation in HepG2 cells. Disabling AhR with pharmacological antagonists or AhR-siRNA abolished the suppressive effects of IS on HIF activation [111].

Hamano et al. showed that IS induces hepcidin production through pathways involving both AhR and oxidative stress in vitro using HepG2 cells. In vivo experiments using adenine-induced CKD mice showed increased hepatic and plasma hepcidin levels, decreased expression of FPN in the duodenum, and increased splenic iron concentration. Plasma ferritin levels were elevated, but plasma iron levels were reduced, suggesting impaired iron absorption and utilization. The removal of IS with AST-120 controlled the increase in hepcidin, improving iron metabolism, promoting iron mobilization, and ameliorating erythropoiesis [112].

EPORs are expressed not only in erythroblasts but also in endothelial cells, vascular smooth muscles, and cardiomyocytes. EPO promotes proliferation, anti-apoptosis, and activation of endothelial nitric oxide synthase (eNOS) in endothelial cells and also stimulates the expression and secretion of the hematopoietic stimulator thrombospondin-1 (TSP-1) [113]. Adelibieke et al. demonstrated that IS inhibits tyrosine phosphorylation of EPORs in endothelial cells, suppressing Akt phosphorylation and thus inhibiting the proliferative, anti-apoptotic, and eNOS-activating effects of EPO, as well as TSP-1 expression [114].

Additionally, IS contributes to eryptosis, known as red blood cell suicide, characterized by erythrocyte shrinkage due to extracellular calcium (Ca^2+^) influx [115]. IS increases intracellular Ca^2+^ in erythrocytes, leading to enhanced phosphatidylserine (PS) exposure and microparticle (MP) release. During apoptosis, PS externalizes to the outer membrane and combines with released MPs, facilitating early detection of apoptotic cells [116]. Dias et al. compared healthy individuals with hemodialysis patients, showing that IS increases the generation of ROS and eryptosis through mechanisms dependent on OAT2, NADPH oxidase activity, and independent of GSH [117]. These findings support the role of IS in the pathophysiology of renal anemia.

Other studies have shown a significant negative correlation between IS levels and EPO expression in CKD patients [118]. For instance, administration of AST-120 (an oral adsorbent that binds IS precursors) to CKD rats led to decreased serum IS levels and increased EPO expression. Conversely, an observational study involving larger patient groups did not find a significant association between IS and anemia [119]. IS undergoes sulfation by SULT1A1, and recent research by Hou, H. using SULT1A1-deficient mice revealed that IS-induced renal fibrogenesis involves enhanced Wnt/β-catenin signaling, inhibition of macrophage infiltration in renal tissues, which normally suppress fibrosis, and decreased EPO production activity. The development of drugs selectively inhibiting SULT1A1 could offer potential for preventing or reducing renal fibrosis and slowing CKD progression [120]. These findings highlight the involvement of IS in the onset and regulation of renal anemia, providing insights into potential therapeutic targets for managing this condition in CKD patients.

### 4.5. Other Protein-Bound Uremic Toxins Involved in Renal Anemia

Apart from IS, several protein-bound uremic toxins have been identified that contribute to renal anemia through mechanisms distinct from EPO production.

In patients with ESKD, polyamines have been reported to act as inhibitors of erythropoiesis, diminishing the proliferation and maturation of red blood cell precursors (colony-forming unit-erythroid) [121]. Acrolein, produced from polyamines, stimulates ceramide formation and induces eryptosis, characterized by erythrocyte contraction due to extracellular Ca^2+^ entry [122]. These studies collectively suggest that not only IS but also other uremic toxins are critical in the development of renal anemia. However, a deeper understanding of the final accumulation mechanisms of these toxins is necessary for more comprehensive research. This insight into the role of various protein-bound uremic toxins in renal anemia underscores the complexity of the condition in CKD patients. It highlights the need for targeted therapeutic strategies that address not only the deficiency in EPO production but also the broader spectrum of metabolic and cellular disturbances caused by these toxins. As research continues to unravel the multifaceted nature of renal anemia, new avenues for effective treatment and management of this condition in CKD patients are likely to emerge.

## 5. Inflammatory Cytokines

### 5.1. Inflammatory Cytokines in End-Stage Kidney Disease

In patients with ESKD, the levels of inflammatory cytokines and chemokines are consistently elevated in circulation. Activated leukocytes in these patients produce ROS, contributing to oxidative stress [123,124]. Inflammatory cytokines fall under the category of middle-molecular-weight uremic toxins, constituting approximately 28% of all identified uremic toxins [80].

TNF-α and IL-6 are prime examples of inflammatory cytokines implicated in mediating inflammation associated with the progression of CKD and cardiovascular disease (CVD). Niewczas et al. identified a kidney risk inflammatory signature (KRIS) consisting of 17 proteins enriched in TNF receptor superfamily members that are associated with a 10-year risk of ESKD. KRIS contributes to the inflammatory process underlying the development of ESKD in diabetes [125].

### 5.2. Tumor Necrosis Factor-Alpha

TNF-α, with a molecular weight of 17.3 kDa, is produced by immune cells like activated T lymphocytes, macrophages, and mast cells, as well as by various other cells such as vascular endothelial cells, renal tubular epithelial cells, mesangial cells, and cardiac myocytes. While circulating levels of TNF-α are negligible or undetectable in healthy conditions, they rise during acute and chronic inflammation [126].

The relationship between TNF-α and renal function has been explored in animal models and in patients at various stages of CKD. In a study of 75 patients with CKD stage 2–5, plasma levels of TNF-α were inversely correlated with GFR and started to accumulate at GFR levels < 81 mL/min/1.73 m^2^ [127]. Experimental evidence suggests that TNF-α contributes to renal vasoconstriction through increased superoxide production, reducing the bioavailability of nitric oxide (NO) and the subsequent decline in GFR [128]. The binding of TNF-α to TNFR1 alters renal hemodynamics, reducing renal blood flow and GFR, while its interaction with TNFR2 triggers macrophage infiltration into the renal interstitium, leading to interstitial fibrosis and glomerulosclerosis [129]. This finding aligns with accumulated evidence that renal inflammation activates immune cells and contributes to fibrosis-promoting mechanisms leading to ESKD [130]. TNF-α contributes to the progression of atherosclerosis. In human umbilical vein endothelial cells, TNF-α binds to TNFR1 and stimulates the expression of vascular cell adhesion molecule-1, intercellular adhesion molecule-1, and E-selectin by causing downstream NF-κB activation, thereby increasing endothelial inflammation and atherosclerosis [131].

### 5.3. Interleukin-6

IL-6, like TNF-α, plays a vital role in both acute and chronic inflammation [132]. It has been shown to exert multifaceted effects on immune responses and hematopoiesis [133]. IL-6 synthesis is induced by other inflammatory cytokines like TNF-α, IL-1α, bacterial lipopolysaccharides, and viral infections [134]. It also triggers various biological activities, such as stimulating fibroblast-like synovial cells to produce receptor activators of the NF-κB ligand, essential for osteoclast differentiation and activation [135]. IL-6 also induces overproduction of vascular endothelial growth factor, leading to enhanced angiogenesis and vascular permeability [136].

Increased plasma IL-6 levels are observed in CKD patients [137], mRNA expression in renal biopsies from human CKD patients has shown an increase in IL-6 compared to controls [138]. Chen et al. demonstrated that the inhibition of IL-6 trans-signaling through the antagonistic action of IgG Fc-linked gp130 on the IL-6/sIL-6R complex reduces STAT3 phosphorylation, thereby inhibiting the development of renal fibrosis [139]. Additionally, the inhibition or genetic absence of IL-6 prevented the deleterious effects of angiotensin II, including hypertension, endothelin-1 expression, and renal injury/fibrosis [140]. The trans-signaling of IL-6 is implicated in the expression of FGF23, a biomarker closely associated with elevated serum levels in CKD patients and strongly correlated with mortality [141].

### 5.4. Impact of Inflammatory Cytokines on Iron Metabolism and Renal Anemia

Inflammatory cytokines have been implicated in the suppression of bone marrow erythropoiesis in CKD patients, contributing to anemia. In a hepcidin-independent pathway, TNF-α causes the relocalization of FPN in intestinal epithelial cells, leading to systemic iron deficiency [142]. IL-6 is a significant factor in hepcidin control [76]. Hepcidin is induced by the activation of STAT3, triggered by IL-6 and other cytokines [70] (Figure 4). The increase in hepcidin may serve as a host defense strategy, limiting the iron necessary for invading microbes or malignant cells. It has been demonstrated that intravenous pre-treatment with hepcidin or induction of hepatic hepcidin expression in a mouse model prevented inflammatory responses to lipopolysaccharide exposure [143]. A report using hepcidin-knockout mice in a turpentine-induced inflammation model has shown that an increase in hepcidin plays a crucial role in the dysregulation of iron homeostasis in inflammatory states [144]. In addition to hepcidin, it has been shown that H-ferritin 1, produced by myeloid cells in response to infection, plays an important role in iron redistribution [145].

The increase in hepcidin expression during inflammation and infection explains the sequestration of iron in macrophages and the inhibition of intestinal iron absorption, displaying two characteristics of anemia of inflammation: normocytic or microcytic hypochromic [146]. This mechanism is believed to be involved in the development of functional iron deficiency and anemia in patients with chronic inflammatory diseases [147,148,149].

## 6. Treatment

### 6.1. Treatment of Anemia in CKD

Effective management of renal anemia involves stimulating red blood cell synthesis and maintaining sufficient levels of iron for Hb formation. The National Institute for Health and Care Excellence (NICE) recommends using either iron or erythropoiesis-stimulating agents (ESAs), or a combination of both, to treat anemia in CKD [150]. This approach targets both absolute and functional iron deficiencies, which can limit iron utilization and lead to EPO deficiency.

### 6.2. Iron Supplementation

Intravenous administration of iron is efficient in replenishing iron stores, promoting erythropoiesis, and reducing the required doses of ESAs. This approach is vital in averting adverse outcomes associated with high ESA doses [151]. Multicenter placebo-controlled randomized controlled trials in patients with pre-dialysis have shown significant improvements in composite endpoints (total mortality, initiation of dialysis, renal transplantation) in groups receiving iron [152]. However, intravenous iron administration has been shown to potentially increase oxidative stress, infection risk, hypersensitivity reactions, and cardiovascular mortality [153,154,155]. It has been suggested that intravenous iron sucrose can increase superoxide production and monocyte adhesion to the endothelium, inducing atheromatous plaque formation [156]. While beneficial for functional iron-deficiency anemia in CKD patients, iron administration can negatively impact markers of oxidative stress and potentially influence clinical outcomes [151], and iron overload should be carefully monitored.

### 6.3. ESA and Its Hyporesponsiveness

EPO deficiency is a primary cause of anemia in CKD; thus, ESAs are employed in treatment. Most CKD patients responsive to ESA therapy exhibit improved anemia and QOL [157]. However, high doses of ESA are associated with increased risks of myocardial infarction, chronic heart failure, stroke, and mortality [158,159].

ESA hyporesponsiveness exists where anemia does not improve despite adequate ESA administration [160]. Approximately 10% of CKD-induced anemia patients show low responsiveness to ESA therapy [157]. Potential causes include chronic bleeding, chronic inflammation, malignancies, iron deficiency, blood disorders, and malnutrition. Hepcidin has been suggested to play a role in ESA resistance. In CKD patients, studies on the relationship between hepcidin levels, ESA dosage, and ESA resistance are inconsistent and debatable. Pinto et al. reported that EPO directly affects hepcidin expression in a dose-dependent manner [161].

It has been reported that hepcidin levels are significantly higher in the ESA hyporesponsive group than in the ESA-sensitive group [162,163]. In contrast, a study involving around 100 hemodialysis patients undergoing ESA maintenance therapy found lower hepcidin values in patients receiving higher ESA doses, regardless of achieved Hb levels [164]. Other studies reported lower serum hepcidin levels in ESA-resistant groups compared to ESA-sensitive groups [165,166], and a significant negative correlation between serum hepcidin levels and weekly EPO dosage [167]. However, another cross-sectional study involving over 400 hemodialysis patients showed no association between hepcidin values and ESA dosage or intravenous iron therapy [168]. In other reports, hepcidin levels in ESA-responsive and ESA-resistant hemodialysis patients were similar [169]. In cardiorenal syndrome patients, responders to ESA showed higher hepcidin-25 levels, suggesting that hepcidin could be more of a marker for iron load and ESA responsiveness than for inflammation or EPO resistance [170].

### 6.4. Other Treatments

Emerging as a novel treatment for anemia is the use of oral HIF prolyl hydroxylase domain inhibitors to stabilize HIF and thereby enhance endogenous EPO secretion [14,171]. Roxadustat, for example, has shown efficacy in stimulating erythropoiesis and regulating iron metabolism in patients with ESKD [172]. AST-120, by absorbing indole in the intestines, lowers serum levels of IS, thereby enhancing its excretion in feces [173]. Administration of AST-120 in conjunction with ESAs has been linked to improved Hb levels [174] (Figure 5).

### 6.5. Targeting Hepcidin as a Therapeutic Strategy

Given the role of inflammation and elevated hepcidin in suppressing erythropoiesis and reducing iron availability, directly targeting hepcidin as a therapeutic strategy has been proposed [175]. The development of anticalins, lipocalins capable of binding hydrophobic low-molecular-weight compounds like hepcidin and inhibiting their function offers a new avenue in CKD anemia treatment. PRS-080, a pegylated anticalin protein, has shown promise in clinical trials, reducing hepcidin and increasing serum iron and Tf saturation [176] (Figure 5). Hepcidin expression inhibitors, such as heparin, have shown potential both in vitro and in vivo [177]. Heparin suppresses hepcidin expression through the BMP6/SMAD pathway, making it a promising candidate for CKD anemia treatment. Pentosan polysulfate, with low anticoagulant activity and high sulfation like heparin, is already clinically used for interstitial cystitis and osteoarthritis and has been shown to reduce serum hepcidin in mouse models [178]. Vitamin D supplementation reduces hepcidin gene transcription, promoting erythropoiesis and reducing inflammation [179]. In early CKD patients, vitamin D3 supplementation reduced hepcidin levels over three months [180]. However, further research is needed to confirm the long-term effects of these novel treatments on CKD patients.

## 7. Summary

In summary, this comprehensive review highlights the intricate interplay between iron metabolism and inflammatory mediators in patients with renal dysfunction. CKD, a prevalent global health issue, is closely associated with disturbances in iron homeostasis and elevated levels of inflammatory cytokines, exacerbating complications like renal anemia and CVD. Renal anemia, primarily stemming from reduced EPO production and iron dysregulation, significantly impacts patient outcomes. Uremic toxins, particularly protein-bound compounds like IS, play a pivotal role in advancing CKD and influencing iron metabolism, leading to complex regulatory challenges in managing renal anemia. Furthermore, inflammatory cytokines like TNF-α and IL-6 are integral to the progression of CKD and the perturbation of iron metabolism. In addition, modern iron status indicators like hepcidin and HJV might be used to assess iron metabolism and predict its course in CKD patients.

Treatment strategies for renal anemia in CKD will have evolved, focusing on addressing both EPO and iron deficiencies and considering the broader spectrum of metabolic disturbances. Novel approaches, like targeting hepcidin and utilizing HIF stabilizers, will have offered promising therapeutic avenues. The intricate connections between iron metabolism, inflammation, and renal dysfunction will have underscored the need for integrated, multifaceted treatment strategies to improve outcomes for patients with CKD.

## Figures and Tables

**Figure 1 ijms-25-03745-f001:**
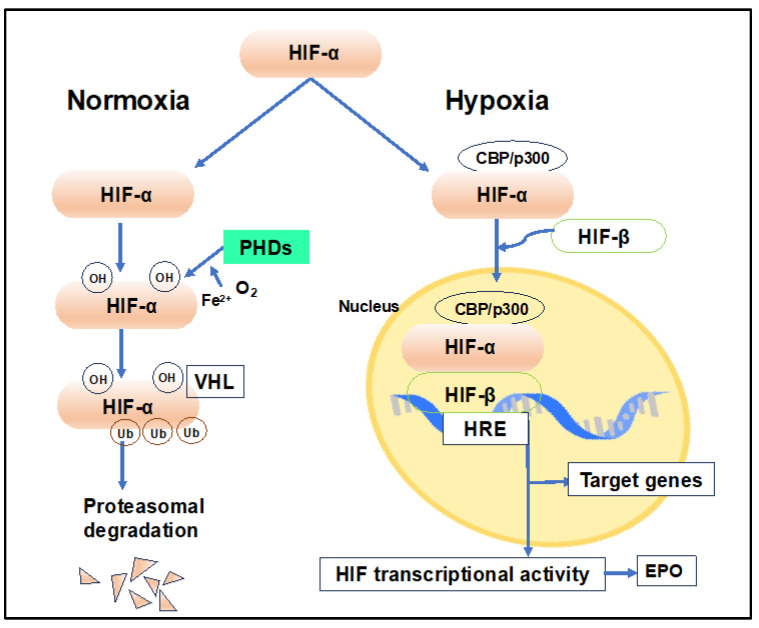
Mechanisms of hypoxic response. Under normal oxygen conditions, HIF-α is degraded by PHD and VHL. Under hypoxic conditions, PHD is inactivated, resulting in HIF not being degraded but instead translocated into the nucleus. There, it forms a HIF-α/β complex that binds to HRE, thus promoting the transcription of EPO. HIF: hypoxia-inducible factor, PHD: prolyl hydroxylase domain enzyme, VHL: Von Hippel-Lindau, HRE: hypoxia responsive element, EPO: erythropoietin.

**Figure 2 ijms-25-03745-f002:**
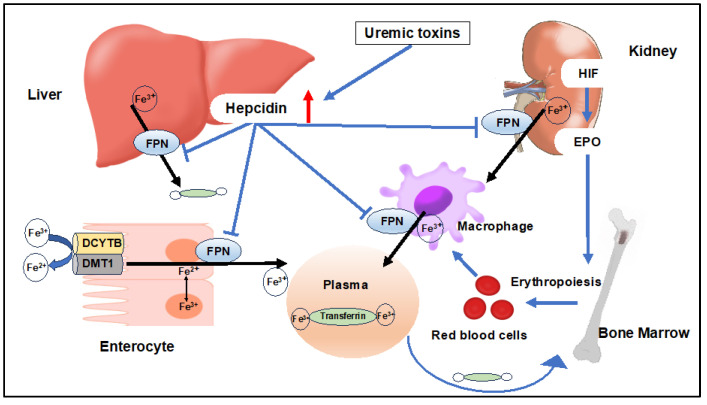
Normal iron metabolism and impaired utilization of iron due to uremic toxins. In the duodenum, ferric iron (Fe^3+^) is reduced to ferrous iron (Fe^2+^) by DCYTB and transported into enterocytes by DMT1. Absorbed iron is exported to the plasma by the FPN, where it binds to Tf for circulation. Circulating Tf-bound iron is transported to the liver and spleen for storage in ferritin or to the bone marrow for erythropoiesis. Aged red blood cells are phagocytized by macrophages and recycling iron from erythrocyte destruction. EPO is produced in the kidney via HIF activation and induces red blood cell production. FPN expression is regulated by hepcidin. Hepcidin regulates the uptake of iron from the intestine and the release of stored iron. Hepcidin is induced by uremic toxins and impairs the utilization of available iron. DCYTB: duodenal cytochrome B, DMT1: divalent metal transporter 1, FPN: ferroportin, Tf: transferrin, HIF: hypoxia-inducible factor, EPO: erythropoietin.

**Figure 3 ijms-25-03745-f003:**
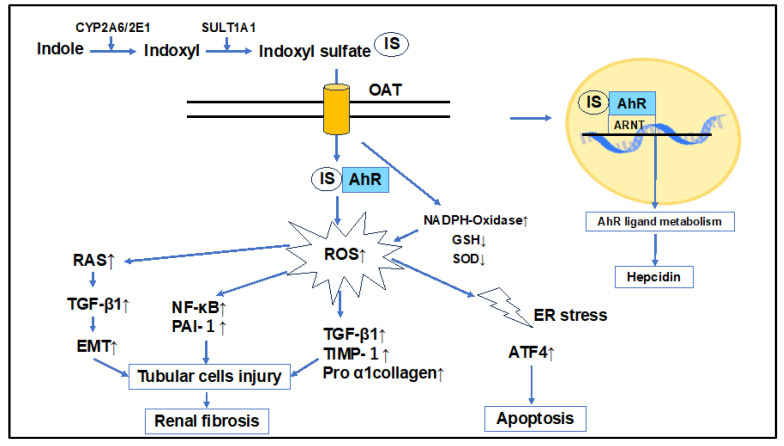
Indoxyl sulfate and its impact on kidney damage. IS is transported through OAT, which leads to increased ROS production, activates various cytokines, including NF-κB (e.g., RAS, TGF-β), and induces EMT in tubular cells, further promoting renal fibrosis. IS enhances ROS production in endothelial cells, activates NADPH oxidase, decreases levels of the active antioxidant GSH, and induces oxidative stress. It also reduces the activity of the renal antioxidant enzyme SOD. IS-related ER stress leads to an increase in ATF4 and thus induces apoptosis. IS acts through AhR, and activation of AhR induces hepcidin production. IS accelerates the progression of glomerulosclerosis, indicating its role in advancing renal failure. IS: indoxyl sulfate, OAT: organic anion transporter, SULT1A1: sulfotransferase enzyme, CYP2A6/2E1: cytochrome P450 enzymes, AhR: aryl hydrocarbon receptor, ROS: reactive oxygen species, GSH: glutathione, SOD: superoxide dismutase, TGF-β1: transforming growth factor-β1, TIMP-1: tissue inhibitor of metalloproteinases 1, EMT: epithelial-to-mesenchymal transition, ER stress: endoplasmic reticulum stress, ATF4: activating transcription factor 4.

**Figure 4 ijms-25-03745-f004:**
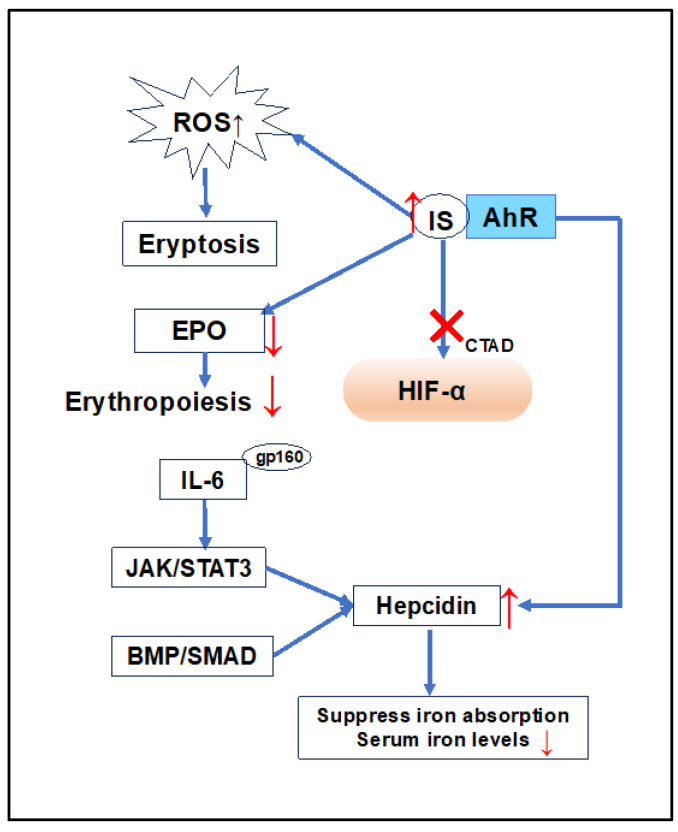
Pathophysiology of renal anemia. As CKD progresses, the accumulation of IS occurs within the body. IS inhibits the hypoxic induction of HIF-1 target genes and disrupts the function of the CTAD of HIF-1α. IS increases the generation of ROS and eryptosis. IS leads to a deficiency in EPO production by renal interstitial fibroblasts, resulting in a decrease in red blood cell production and anemia. Activation of AhR induces hepcidin production and inhibits activation of HIF. Hepcidin is induced by the activation of STAT3, triggered by IL-6 and other cytokines. Furthermore, the expression of hepcidin is induced by the activation of the BMP/SMAD pathway. IS: indoxyl sulfate, HIF: hypoxia-inducible factor, EPO: erythropoietin, CTAD: C-terminal transactivation domain, AhR: aryl hydrocarbon receptor, ROS: reactive oxygen species, JAK/STAT: JAK/STAT pathway, BMP/SMAD: BMP/SMAD pathway.

**Figure 5 ijms-25-03745-f005:**
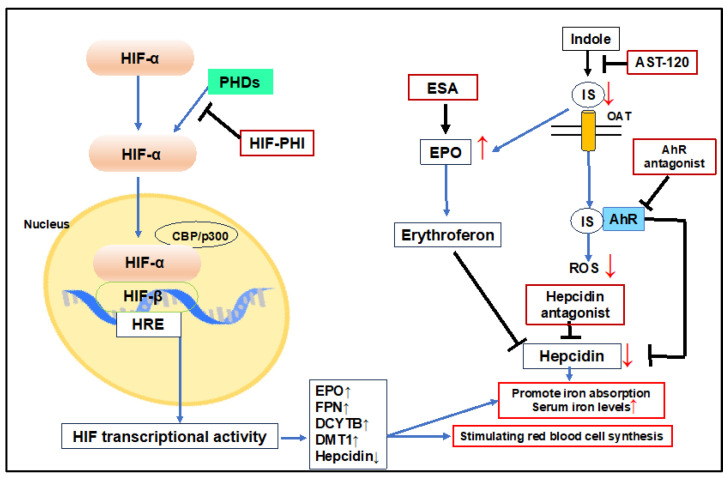
Treatment of renal anemia. HIF-PHI inhibits PHD activity, stabilizes HIF-α, and promotes endogenous EPO production. Activation of HIF induces expression of FPN, DCYTB, and DMT1 and promotes iron utilization. Hematopoietic stimulation by EPO suppresses hepcidin production via erythroferon. AST-120 decreases IS levels and increases EPO expression. Decreasing IS levels reduces ROS generation. The AhR antagonist inactivates AhR and regulates hepcidin increase. Hepcidin antagonists decrease hepcidin and promote iron utilization. This approach improves anemia by promoting iron utilization and erythropoiesis. HIF: hypoxia-inducible factor, PHD: prolyl hydroxylase domain enzyme, EPO: erythropoietin, HRE: hypoxia responsive element, HIF-PHI: HIF prolyl hydroxylase domain inhibitors, ESA: erythropoiesis-stimulating agent, FPN: ferroportin, DCYTB: duodenal cytochrome B, DMT1: divalent metal transporter 1, AhR: aryl hydrocarbon receptor, IS: indoxyl sulfate.

## Data Availability

Data are contained within the article.

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
