# Peer review of "Iron Metabolism and Inflammatory Mediators in Patients with Renal Dysfunction"

_ijms, 2024, doi:10.3390/ijms25073745_

Round 1

Reviewer 1 Report

Comments and Suggestions for Authors

An interesting, well-written manuscript by Japanese authors on the role of parameters describing iron metabolism and inflammatory mediators in patients with kidney diseases.

Below I am attaching comments to the manuscript:

 1. there are minor stylistic errors in the content of the paper, but they do not detract from the great value of the manuscript.

2. I do not have any major comments regarding the content of the paper because it is very detailed and meticulous. However, the authors could describe in perspective the possible use of other parameters of iron metabolism that may be used in the diagnosis or prediction of the course of kidney diseases, I am thinking here, first of all, about hemojuvelin.

3. For this reason, I would suggest replacing the summary point with future perfectives and, of course, using the text from the previous chapter, but also adding what other parameters may be useful in the assessment of iron metabolism in patients with kidney diseases. The authors can use publications that do not concern kidney diseases themselves, but concern new parameters of iron metabolism in human physiopathology. I propose the following manuscripts:

a. doi: 10.3390/cancers15041041.

b. doi: 10.3390/ijms23010269.

c. doi: 10.3390/cells9122591.

Author Response

Responses to Comments Resulting from the Editorial Conference:

Thank you very much for providing important comments. We are thankful for the time and energy you expended. We have made some revisions to the content, in addition to some that were pointed out to me by the reviewers.

Reviewer 1's corrections are indicated in blue in the text, Reviewer 2's corrections are in red.

Our responses to the referees’ comments are as follow:

  • Reviewer 1

An interesting, well-written manuscript by Japanese authors on the role of parameters describing iron metabolism and inflammatory mediators in patients with kidney diseases.

Below I am attaching comments to the manuscript:

1). there are minor stylistic errors in the content of the paper, but they do not detract from the great value of the manuscript.

2). I do not have any major comments regarding the content of the paper because it is very detailed and meticulous. However, the authors could describe in perspective the possible use of other parameters of iron metabolism that may be used in the diagnosis or prediction of the course of kidney diseases, I am thinking here, first of all, about hemojuvelin.

3). For this reason, I would suggest replacing the summary point with future perfectives and, of course, using the text from the previous chapter, but also adding what other parameters may be useful in the assessment of iron metabolism in patients with kidney diseases. The authors can use publications that do not concern kidney diseases themselves, but concern new parameters of iron metabolism in human physiopathology. I propose the following manuscripts:

  1. doi: 10.3390/cancers15041041.
  2. doi: 10.3390/ijms23010269.
  3. doi: 10.3390/cells9122591.

→Thank you very much for providing important comments.

Within section 3-5, there is a section describing factors that regulate hepcidin expression, and we added there a description of the BMP/SMAD pathway and hemojuvelin as one of the hepcidin regulators. We mentioned there the reference “a” that you provided. We did not add the content of reference “c” due to the volume of the original text, but we added reference “b” in section 5-4 as an explanation of iron redistribution during infection and inflammation.

Additional text is provided in blue within sections 3-5 and 5-4 of the text.

・Section 3−5

“In conditions of iron overload, the expression of hepcidin is induced by the activation of the bone morphogenetic protein (BMP)/SMAD pathway (the human homolog of Drosophila mothers against decapentaplegic). Hemojuvelin (HJV), a membrane protein, is implicated in the iron overload condition known as juvenile hemochromatosis. Serving as a BMP co-receptor, HJV signals through the SMAD pathway to regulate hepcidin expression [77]. Although the study by Styczynski, J. et al. did not focus on patients with CKD, it revealed that soluble HJV (sHJV) plays a role in the regulation of iron metabolism. Furthermore, sHJV could be a valuable biomarker for assessing short-term outcomes in pediatric patients after hematopoietic cell transplantation or chemotherapy, both of which are contexts where iron overload is a concern [78].”

・Section 5−4

“In addition to hepcidin, it has been shown that H-ferritin 1, produced by myeloid cells in response to infection, plays an important role in iron redistribution [148].”

We replaced the summary point with future perfective. We added a description of the potential use of other parameters of iron metabolism that may be used to diagnose and predict the course of kidney disease, as follows in the summary:

“In addition, modern iron status indicators like hepcidin and HJV might be used to assess iron metabolism and predict its course in CKD patients.”

Reviewer 2 Report

Comments and Suggestions for Authors

The manuscript "Iron metabolism and inflammatory mediators in patients with renal dysfunction" is a broad review addressing the complex role of erythropoietin, iron metabolism, inflammation, and uremia in developing anemia and cardiovascular complications in CKD patients. The authors describe multiple converging mechanisms, current challenges in managing anemia, and treatment strategies, including novel approaches such as hepcidin targeting and HIF stabilizers to improve outcomes in CKD patients. The review approaches a relevant subject in CKD; here are some suggestions to improve it:

In section 2-2. Erythropoietin as a Central Factor in Erythropoiesis, the authors introduce the concept of hypoxia and its importance in erythropoietin (EPO) production. The authors focus solely on renal EPO production without mentioning the liver. Although anemia in CKD is primarily related to impaired production of renal EPO, the authors should include a more comprehensive explanation of how and where EPO is produced. Also, in this section, the authors introduce the concept of hypoxia and define HIF as the transcription factor hypoxia-inducible factor. Yet, a more detailed description of the characteristics of hypoxia-inducible factors and the existence of HIF-1 and HIF-2 is also needed. Finally, in this section, the authors focus solely on renal EPO-producing cells and HIF2; what about other mechanisms besides pO2 critical for renal EPO synthesis? The role of HIF1 is highly relevant for hypoxia and EPO production, yet HIF1 is not mentioned. 

Figure legends appear to be part of the main text. Please format them as figure legends.

In Figure 1, the authors should change Fe when depicted in the liver, kidney or bound to transferrin for either a more specific symbol that includes its oxidation state, or write iron.

Comments on the Quality of English Language

Several spacing errors need to be fixed, such as:

Line 46: life(QOL) -> life (QOL)

Line 67: conditions[15] -> conditions [15]; EPO receptors(EPOR) -> EPO receptors (EPOR)

Line 69: differentiation[16][17] -> differentiation [16][17]

Line 71: EPO[18] -> EPO [18]

Line 113: 1-2mg -> 1-2 mg

Several abbreviations appear without being defined when first mentioned, such as:

Line 90: BUN

Line 146: TBI

Line 154: NTBI

Line 270: RAAS

Line 336: OAT2

Line 400: VCAM1, ICAM1

Line 530: BMP6/SMAD

Also, consider if it is worth abbreviating when the terms are used only once.

Line 284: remove endoplasmic reticulum and leave ER since it was previously defined.

Abbreviations are mostly all in caps, except transferrin (Tf) and Scara5 (not defined).

In line 139 if FPN FPN1?

Several superscripts need to be fixed, for example:

Lines 122-123: Fe3+, Fe2+, same for Ca2+

Lines 55-57, 391: 1.73 m2 

Revise the format of reference numbers throughout the manuscript; for example:

Line 69: [16][17] -> [16,17]

Line 196: [30][72][73][74][75][76] -> [30, 72-76]

Line 225: [79][80][81][82][83[ -> [79-83]

Author Response

Responses to Comments Resulting from the Editorial Conference:

Thank you very much for providing important comments. We are thankful for the time and energy you expended. We have made some revisions to the content, in addition to some that were pointed out to me by the reviewers.

Reviewer 1's corrections are indicated in blue in the text, Reviewer 2's corrections are in red.

Our responses to the referees’ comments are as follow:

  • Reviewer 2

The manuscript "Iron metabolism and inflammatory mediators in patients with renal dysfunction" is a broad review addressing the complex role of erythropoietin, iron metabolism, inflammation, and uremia in developing anemia and cardiovascular complications in CKD patients. The authors describe multiple converging mechanisms, current challenges in managing anemia, and treatment strategies, including novel approaches such as hepcidin targeting and HIF stabilizers to improve outcomes in CKD patients. The review approaches a relevant subject in CKD; here are some suggestions to improve it:

In section 2-2. Erythropoietin as a Central Factor in Erythropoiesis, the authors introduce the concept of hypoxia and its importance in erythropoietin (EPO) production. The authors focus solely on renal EPO production without mentioning the liver. Although anemia in CKD is primarily related to impaired production of renal EPO, the authors should include a more comprehensive explanation of how and where EPO is produced. Also, in this section, the authors introduce the concept of hypoxia and define HIF as the transcription factor hypoxia-inducible factor. Yet, a more detailed description of the characteristics of hypoxia-inducible factors and the existence of HIF-1 and HIF-2 is also needed. Finally, in this section, the authors focus solely on renal EPO-producing cells and HIF2; what about other mechanisms besides pO2 critical for renal EPO synthesis? The role of HIF1 is highly relevant for hypoxia and EPO production, yet HIF1 is not mentioned. 

→Thank you very much for providing important comments.

We have added a more comprehensive description of EPO production, mentioning production from the liver as well as the kidney, and including a section on the HIF active pathway. Section 2-2 was revised to include the following:

2-2. Erythropoietin as a Central Factor in Erythropoiesis

EPO is a critical glycoprotein hormone for erythropoiesis, and its expression increases in hypoxic and anemic conditions. It circulates in the plasma and stimulates marrow progenitors, thereby increasing red cell production [8]. It achieves this by binding to EPO receptors (EPOR) on erythroid progenitor cells in the bone marrow, thereby inhibiting apoptosis and stimulating both proliferation and differentiation [9]. During late fetal development, erythropoiesis is supported by EPO production from hepatocytes, but in adults, EPO production primarily occurs in the kidneys. EPO is produced by renal erythropoietin-producing (REP) cells in the proximal peritubular interstitium of the kidney [10].  Furthermore, it has been shown that REP cells originate from myelin protein zero cells in the neural crest [11]. As CKD progresses, REP cells undergo phenotypic transition to myofibroblasts, during which they lose their ability to synthesize EPO [12].

The main determinant of EPO production is the transcriptional activity of its gene in the kidneys, which is related to local oxygen tensions. Thus, EPO production is regulated by hypoxia-inducible factors (HIFs). HIF is a transcription factor consisting of a dimer of HIF-α and HIF-β. HIF-α has three isoforms: HIF-1α, HIF-2α, and HIF-3α. In the kidney, HIF-1α is mainly located in tubular epithelial cells, while HIF-2α is found in endothelial cells and fibroblasts [13]. Under normal oxygen conditions, HIF-1α and HIF-2α undergo hydroxylation of proline residues by prolyl hydroxylases (PHD), leading to ubiquitination by von Hippel-Lindau (VHL) protein and subsequent degradation by the proteasome. On the other hand, under hypoxic conditions, reduced activity of PHD stabilizes HIF-α proteins, forming a dimer with HIF-β and binding to hypoxia response elements (HREs), inducing the expression of target genes such as EPO [Figure 1].

Although the kidney is the primary site of EPO production associated with HIF activation, EPO production also originates from the liver, as low levels of EPO production were observed with PHD inhibitors in CKD patients after nephrectomy [14]. PHD also has three isoforms, and while in the liver, all PHD isoforms must be inhibited [15, 16], in the kidney, PHD2 is said to be most involved [17]. HIF activation influences not only EPO production but also genes that play important roles in iron metabolism, including transferrin (Tf) [18], ferroportin (FPN) [19], and possibly hepcidin [19].

Figure legends appear to be part of the main text. Please format them as figure legends.

In Figure 1, the authors should change Fe when depicted in the liver, kidney or bound to transferrin for either a more specific symbol that includes its oxidation state, or write iron.

→We aligned the format of figure legends to the same as the main format. Considering the order of explanation, the original Figure 1 has been changed to Figure 2. We showed the oxidation state of iron bound to liver, kidney, or transferrin in Figure 2.

Comments on the Quality of English Language

Several spacing errors need to be fixed, such as:

Line 46: life(QOL) -> life (QOL)

Line 67: conditions[15] -> conditions [15]; EPO receptors(EPOR) -> EPO receptors (EPOR)

Line 69: differentiation[16][17] -> differentiation [16][17]

Line 71: EPO[18] -> EPO [18]

Line 113: 1-2mg -> 1-2 mg

Several abbreviations appear without being defined when first mentioned, such as:

Line 90: BUN

Line 146: TBI

Line 154: NTBI

Line 270: RAAS

Line 336: OAT2

Line 400: VCAM1, ICAM1

Line 530: BMP6/SMAD.  

Also, consider if it is worth abbreviating when the terms are used only once.

Line 284: remove endoplasmic reticulum and leave ER since it was previously defined.

Abbreviations are mostly all in caps, except transferrin (Tf) and Scara5 (not defined).

In line 139 if FPN FPN1?

Several superscripts need to be fixed, for example:

Lines 122-123: Fe3+, Fe2+, same for Ca2+

Lines 55-57, 391: 1.73 m2 

Revise the format of reference numbers throughout the manuscript; for example:

Line 69: [16][17] -> [16,17]

Line 196: [30][72][73][74][75][76] -> [30, 72-76]

Line 225: [79][80][81][82][83[ -> [79-83]

→We have corrected the sections you noted. Modifications are noted in red letters.

We defined organic anion transporters as OATs in line 282, so OAT2(in the new paper, line 367) is now an abbreviation. We defined the BMP/SMAD pathway on line 225, so we omitted the definition on line 530 (559 in the new paper).

  • Other Corrections

・Figure originally labeled Figure 3 was split and part of Figure 3 was changed to Figure 1

・Added text to Section 3-5, lines 220-222.(The original text is in Section 5-3):

“Furthermore, HIFs, through the stimulation of EPO-induced erythropoiesis, indirectly reduce serum levels of hepcidin [75]”

・Deleted the following sentence from the Introduction:

“A Japanese cohort study underscored the importance of etiological identification in CKD, particularly in diabetic nephropathy, noting significant differences in patient outcomes when compared to other renal diseases.”

・Deleted the last sentence of section 5-3:

“IL-6 is considered a primary biomarker for cardiovascular risk in CKD”

・Revised the description of section 5-4 and section6-2, and section6-5. (Yellow highlighting in the text.)

Round 2

Reviewer 1 Report

Comments and Suggestions for Authors

I think that the authors have adequately addressed the comments made by the reviewers.

Reviewer 2 Report

Comments and Suggestions for Authors

The authors made significant changes to their manuscript and addressed all the issues raised during the first review round.